# The Perfusion Index of the Ear as a Predictor of Hypotension Following the Induction of Anesthesia in Patients with Hypertension: A Prospective Observational Study

**DOI:** 10.3390/jcm11216342

**Published:** 2022-10-27

**Authors:** Ji Young Min, Hyun Jae Chang, Su Jung Chu, Mee Young Chung

**Affiliations:** 1Department of Anesthesiology and Pain Medicine, Eunpyeong St. Mary’s Hospital, College of Medicine, The Catholic University of Korea, 1021 Tongil-ro, Eunpyeong-gu, Seoul 03312, Korea; 2Department of Anesthesiology and Pain Medicine, T. Vincent’s Hospital, College of Medicine, The Catholic University of Korea, 93 Jungbu-daero, Paldal-gu, Suwon-si 16247, Gyeonggi-do, Korea

**Keywords:** anesthesia, hypertension, hypotension, induction, perfusion index, vascular tone

## Abstract

Patients with hypertension develop hemodynamic instability more frequently during anesthesia—particularly post-induction. Therefore, different monitoring methods may be required in patients with hypertension. Perfusion index—the ratio of the pulsatile blood flow to the non-pulsatile static blood flow in a patient’s peripheral tissues, such as the fingers or ears—can show the hemodynamic status of the patient in a non-invasive way. Among the sites used for measuring the perfusion index, it is assumed that the ear is more reliable than the finger for hemodynamic monitoring, because proximity to the brain ensures appropriate perfusion. We hypothesized that the low value of preoperative ear PI could be a predictor of post-induction hypotension in patients with hypertension. Thirty patients with hypertension were enrolled. The perfusion index and pleth variability index were measured using the ear, finger, and blood pressure, and heart rate was recorded to monitor hypotension. After insertion of the supraglottic airway, 20 patients developed post-induction hypotension. Those who developed hypotension showed a significantly lower preoperative perfusion index of the ear. The preoperative perfusion index of the ear could predict post-induction hypotension in patients with hypertension.

## 1. Introduction

Hemodynamic depression, such as hypotension, is common in patients under general anesthesia. Hypotension causes organ hypo-perfusion, leading to ischemia in the major organs, and is an independent contributor to complications such as cardiovascular ischemia or cerebral infarction, heart failure, kidney damage, increased hospital stays, and increased mortality within a year [1,2,3,4]. In particular, hypotension occurs more frequently during the post-induction period. This period refers to the time after the administration of many anesthetic drugs that provoke vascular dilatation and cardiovascular suppression [5,6,7]. 

Patients with hypertension develop more hemodynamic instability than those without hypertension. According to Charlson et al. [8,9], patients with hypertension are exposed to hemodynamic instability during anesthesia, which is known to be associated with adverse perioperative outcomes such as acute renal failure or cerebral ischemia. Patients with hypertension may develop significant hypotension after the induction of anesthesia when the initial sympathetic nervous system response to laryngoscopy and endotracheal intubation—which typically increases blood pressure—is subsided.

The cause of this hemodynamic instability is that patients with hypertension are exposed to high blood pressure for a long time, leading to anatomical or pathophysiological changes in the hemodynamic systems. These changes make them sensitive to the effects of anesthetic drugs that frequently cause hypotension [10,11,12]. Determining hemodynamic state provides a diagnostic and treatment basis for preventing systemic target organ damage, facilitating customized care for patients with hypertension [13]. From this point of view, monitoring preoperative hemodynamic systems can be considered a cardinal aspect of the management of patients with hypertension.

The detailed hemodynamic values are monitored in invasive ways, such as by using a Swan–Ganz catheter. Echocardiography takes time and is costly. Recently, non-invasive methods for determining patients’ hemodynamic status have been used. Among them, the perfusion index (PI) is the ratio of the pulsatile blood flow to the non-pulsatile static blood flow in a patient’s peripheral tissues—such as the ears or fingers—determined via plethysmography. PI reflects the patient’s hemodynamic state [14,15]. However, monitoring using plethysmography varies dramatically depending on the measurement site. The performance of sensors placed on the fingers can be compromised by several conditions, such as a low-perfusion state or positive-pressure ventilation, which can decrease the utility of an in-finger pulse oximeter under crucial conditions [16]. In comparison, the ear guarantees more reliable information than the fingers in terms of plethysmography, because the proximity to the brain ensures appropriate perfusion conditions, as it is supplied directly by a network of arteries from the vascular system to supply the brain [17]. Therefore, the ear may be more suitable than the peripheral locations for monitoring the underlying hemodynamic status [17,18,19,20,21]. 

We hypothesized that PI measured using an ear probe could represent the preoperative hemodynamic status in patients with hypertension and could be useful to predict post-induction hypotension. This prospective study investigated the correlation between post-induction hypotension and PI of the ear measured during the preoperative period in patients with hypertension. 

## 2. Materials and Methods

### 2.1. Study Population

The present study was registered with the Institutional Review Board (IRB) and the Hospital Research Ethics Committee of The Catholic University of Korea, Eunpyeong, St. Mary’s Hospital (IRB protocol no. PC22OISI0006) and registered with the Clinical Research Information Service (CRIS; https://cris.nih.go.kr, KCT0007172, accessed on 7 April 2020). Written informed consent was obtained from each patient before enrollment after approval of the IRB. A total of 30 patients (aged 40–75 years) classified by the American Society of Anesthesiologists as having physical status II with hypertension and who underwent elective surgery between April and July 2022 were enrolled. The exclusion criteria were left ventricular ejection fraction < 40% if transthoracic echocardiography was performed, pre-existing severe vascular disease, implanted pacemaker, autonomic nervous system impairment, unstable vital signs, and expected difficulty in the airway. 

### 2.2. Anesthesia and Hemodynamic Monitoring 

Upon arrival at the operating room, the patients were monitored via electrocardiography. Systolic blood pressure (SBP), diastolic blood pressure (DBP), mean blood pressure (MBP), heart rate (HR), and oxygen saturation (SpO2) were measured using non-invasive methods. The E1 Ear Sensor (Masimo Corp., Irvine, CA, USA) was attached to a concha of the ear, and the rainbow sensor SET^TM^ (Masimo Corp., Irvine, CA, USA) was attached to the middle or index finger of the dominant arm of each patient to continuously monitor the PI and pleth variability index (PVI). PI was defined as the ratio of the pulsatile blood flow to the non-pulsatile static blood flow in a patient’s peripheral tissue, such as an ear or finger. PVI was defined as a continuous non-invasive measure of the relative variability in the plethysmography (pleth) during respiratory cycles. The patient state index was measured using a SedLine^®^ electroencephalograph sensor (Masimo Corp., Irvine, CA, USA) for anesthetic depth. Before anesthesia, the PI and PVI of the ear and finger, along with the SBP, DBP, MBP, HR, and SpO2 were measured and recorded for baseline values. Simultaneously, propofol (Fresofol^®^ MCT 1%; Fresenius Kabi Austria GmbH, Graz, Austria) infusion was initiated with a target effect-site concentration of 3.0 µg/mL using the Marsh pharmacokinetic model. The target effect-site concentration of propofol was changed to 0.0 µg/mL after confirmation of loss of consciousness. Subsequently, the propofol infusion was stopped, and the dose of propofol used for induction was recorded. Concurrently, to facilitate an end-tidal sevoflurane concentration increase, a supra-minimal alveolar concentration (MAC) dose of approximately 1.3 MAC was administered by bag-mask ventilation using a high fresh gas flow (8 L/min) to achieve 1.0 MAC. After administration of 1.2 mg/kg rocuronium and 40 µg of remifentanil bolus, a supraglottic airway device (i-gel^®^; Intersurgical Ltd., Wokingham, UK) was inserted according to the manufacturer’s recommendations. After the induction of anesthesia, controlled ventilation was maintained with a fresh gas flow rate of 4 L/min. Sevoflurane was maintained at 1.0 MAC (age-adjusted). Mechanical ventilation was performed with an air–oxygen mixture (fraction of inspired oxygen = 0.5) at an 8 mL/kg tidal volume by calculating the ideal body weight. The leak volume was kept to a minimum during ventilation to minimize the leakage of anesthetic gas by monitoring the spirometry of the ventilator. The PI, PVI of the ear and finger, SBP, DBP, MBP, HR, and SpO2 were recorded at 2.5-min intervals to monitor the occurrence of hypotension from baseline. The time when the lowest SBP occurred was recorded. SBP < 90 mmHg was treated with a rapid intravenous fluid administration (10 mL/kg) or a 5 mg bolus of ephedrine if hypotension persisted. Bradycardia was defined as an HR < 50 bpm and was treated with 0.5 mg intravenous boluses of atropine.

### 2.3. Data Collection

The baseline referred to the patient’s status just before the start time of propofol infusion. The post-induction period was defined as the period during the first 20 min after the induction of anesthesia (i.e., the propofol infusion start time). Post-induction hypotension was defined as SBP < 90 mmHg occurring during the post-induction period. The PI, PVI of the ear and finger, SBP, DBP, MBP, HR, and SpO2 were collected at 2.5 min intervals from baseline, for a total duration of 20 min.

### 2.4. Statistical Analyses 

In a previous study, the area under the curve (AUC) was 0.816 (95% confidence interval (CI), 0.699–0.933; *p* < 0.001) for the predictive power of PI for hypotension following anesthesia [22]. Based on these data, when the number of participants was calculated using the AUC curve, it was determined to be 24. Thirty participants were recruited, considering a loss rate of 20%. R language version 3.3.3 (R Foundation for Statistical Computing, Vienna, Austria) and the T&F program ver. 3.0 (YooJin BioSoft, Goyang, Korea) were used for all statistical analyses. Data are presented as medians (interquartile ranges) for continuous variables. The Wilcoxon rank-sum test was performed to compare the mean differences between the groups with and without hypotension. Data are presented as sample numbers, percentages for categorical variables, and N (%). The correlation between the lowest SBPs and the hemodynamic variables was analyzed using Spearman’s correlation coefficient. The effects of hemodynamic variables on the lowest SBP were investigated using linear regression analysis. A receiver operating characteristic (ROC) curve analysis was performed to analyze the predictive performance of the hemodynamic variables for the occurrence of hypotension. The AUCs among the hemodynamic variables measured from the baseline were compared. The significance level was set at a *p*-value < 0.05.

## 3. Results

### 3.1. Patient Demographics

Thirty patients were enrolled, of whom twenty (66.7%) developed post-induction hypotension. The patients’ characteristics are presented in Table 1. There were six patients with diabetes mellitus in the hypotension group. In the non-hypotension group, four patients had diabetes mellitus. All patients took calcium channel blockers as anti-hypertensive medication on the day of surgery. There were no differences between the non-hypotension and hypotension groups in terms of demographics. 

### 3.2. Hemodynamic Variables

Most of the hypotension occurred at 15 min from baseline in the hypotension group. The PI of the finger, baseline PVI of the ear and finger, SBP, DBP, MBP, and HR were comparable between the hypotension and non-hypotension groups. The baseline PI of the ear in the hypotension group was 0.49 (0.36–0.6), whereas in the non-hypotension group it was 1.55 (1.25–3.03) (*p* < 0.001, Table 2). The PI of the ear in the hypotension group remained lower than that of the finger during the post-induction period (Appendix A). In all patients, the PI of the ear showed less variation during the post-induction period than the PI of the finger (Figure 1A,B). The lowest SBP in the post-induction period was highly correlated with the ear’s baseline PI in all patients, representing the rho = 0.839 (*p* < 0.001) (Table 3), and showed a linear correlation with the ear’s baseline PI in the regression analysis (Figure 2). The regression equation between the two variables was as follows: the lowest SBP = 83.901 + 8.41 × baseline PI of the ear. 

### 3.3. Prediction of the Post-Induction Hypotension

The AUC, sensitivity, specificity, and cutoff values of the preoperative hemodynamic variables are shown in Table 4. Above all, the baseline PI of the ear revealed a high predictive ability for post-induction hypotension, with a sensitivity and specificity of 1.0 (Figure 3).

## 4. Discussion

To the best of our knowledge, the present study is the first to identify a non-invasive measurement that could predict hypotension after the induction of anesthesia in patients with hypertension. The main finding of this prospective observational study was the correlation between the preoperative PI measurement of the ear and a decrease in SBP in patients with hypertension undergoing the induction of general anesthesia. We found that a PI of the ear ≤0.96 in the preoperative period could be used to predict post-induction hypotension with higher sensitivity and specificity. 

The ear was relatively immune to vasoconstrictive challenges, making ear plethysmograph waveforms suitable for monitoring central hemodynamic status [23,24,25]. In addition, proximity to the brain ensures adequate perfusion by arteries from the vascular system to supply the brain [26]. While shock-induced centralization inhibits peripheral pulse oximetry due to peripheral perfusion deficiency, in-ear pulse oximetry is unlikely to be affected [23,24,27]. Consistent with the present study, the PI of the ear showed less variation than that of the finger (Figure 1A, B). This result supports the hypothesis that the environment has less influence on in-ear pulse oximetry. For this reason, PI determined from the plethysmograph ear waveform provides reliable data about hemodynamic status.

PI, measured from the plethysmograph signal of the pulse oximeter, is calculated as the ratio between the pulsatile component (i.e., arterial compartment) and the non-pulsatile component (i.e., venous and capillary blood and other tissues) of the light reaching the detector of the pulse oximeter, used to monitor vascular tone. PI increases with vasodilation and decreases with vasoconstriction [28,29]. Lower PI means increased vascular tone. PI can also represent the stroke volume or cardiac output. The relationship between PI and stroke volume was investigated in previous studies. One previous study investigated the relationship between body position and perfusion index; the PI was lowest during 45-degree sitting in a supine position [30]. A significant correlation between passive foot lifting and PI changes and volume status after fluid loading was demonstrated in recent research [31]. Lian et al. [32] suggested that the PI and cardiac index were highly correlated during the early treatment phase of septic shock. Furthermore, PI was affected by the vascular tone and SV variation in van Genderen ME et al.’s previous research [33]. 

The baseline PI of the ear was significantly lower in the hypotension group. The preoperative vascular tone and volume status are factors in the development of post-induction hypotension in response to anesthetic vasodilation. Because the patients did not undergo endotracheal intubation, hypotension was not compensated by laryngoscopic stimulation. Accordingly, anesthetic vasodilation was the main cause of hypotension [34,35,36]. Patients with hypertension are exposed to high blood pressure for longer, and their preoperative hemodynamic status differs from that of those without hypertension. Hemodynamic changes in hypertension are characterized by the increased vascular tone and reduced stroke volume [37,38]. In addition, the baroreceptor reflex system, which is the compensation mechanism for hypotension, is impaired [39,40,41]. Hemodynamic changes such as high vascular tone and low stroke volume may be demonstrated by low PI of the ear, as was observed in the hypotension group in the present study. In other words, we deduced that these patients might already have a more vasoconstrictive state and lower stroke volume as a preoperative condition. To confirm this speculation, other possibilities related to left ventricular remodeling associated with hypertension must be considered. 

During the post-induction period, the PI of the ear and finger was decreased in the non-hypotension group. Still, the hypotension group maintained a lower value in the PI of the ear. The PI of the finger in the hypotension group was less decreased than in the non-hypotension group. This result may indicate that anesthetic vasodilation was not well compensated in the hypotension group. Vasoconstriction must occur to overcome vasodilation. It may be more challenging to increase vascular tone in the hypotension group. It could be assumed that the baroreceptor reflex system—a mechanism for increasing vascular tone or stroke volume—did not work well. These preoperative statuses may be contributors to post-induction hypotension. Transthoracic echocardiography (TTE) can similarly detect the preoperative hemodynamic state in a non-invasive way. For example, echocardiographic left ventricular hypertrophy—a common finding in patients with hypertension—increases the incidence of cardiovascular events by 2.17-fold [42,43]. Thus, the results of the present study and TTE findings could be correlated if examined.

We found a correlation between the baseline PI of the ear and the lowest SBP in the post-induction period for all participants. Recently, it was demonstrated that PI not only tracks systemic hemodynamic changes but could also be an early predictor of central hypovolemia and compensatory sympathetic activation in healthy volunteers. Højlund et al. [44] demonstrated in their study that mean arterial pressure was highly correlated (rho [95% CI] = 0.9 [0.7–1], *p* < 0.001) with the PI of the finger, and displayed simultaneous tracings of the mean arterial pressure and PI during general anesthesia. The pathophysiology of changes in hypertension over time may also have been exhibited in the present study’s high correlation between the PI of the ear and SBP. 

The two groups showed a high PI (>3.0) of the finger, contrary to a previous result showing a PI < 1.05 of the finger [22]. There was no significant difference in the PI of the finger between the two groups. The present study was conducted on patients with hypertension. Pathophysiological changes caused by long-term exposure to high blood pressure may have contributed to the differences from previous studies. In addition, all of the participants were taking anti-hypertensive medications such as calcium channel blockers, which are vasodilators. The effects of the antihypertensive medication may have also affected the results.

The present study has several limitations. First, all patients were admitted to the operating room after taking antihypertensive drugs on the day of surgery. These medications were calcium channel blockers, not angiotensin receptor blockers (ARBs) or angiotensin-converting enzyme inhibitors (ACEis). ARBs and ACEis cause potent inhibition of the sympathetic nervous system and the renin–angiotensin–aldosterone system (RAAS). More severe hypotension may occur and may not be compensated well. For this reason, it is common practice to skip the ACEi or ARB on the day of surgery; thus, there may have been a different effect on the vascular system for those patients who took calcium channel blockers in the present study. Second, post-induction hypotension was defined as SBP < 90 mmHg. According to the demographic information of the patients, the average age of the patients was 66 (63.5–72). The standard definition of hypotension would be different in the elderly group [6,45]. Consequently, post-induction hypotension was defined as SBP < 90 mmHg. Third, the opioid was used only at the beginning of induction—not during the post-induction period. As the continuous infusion of opioids is common in anesthesia, the findings of the present study are limited. Finally, some patients had diabetes mellitus and hypertension. The perfusion or structure of the vascular system is altered in diabetes mellitus, which may have affected the results of the present study.

## 5. Conclusions

The low value of the preoperative PI of the ear before the induction of anesthesia can be used to predict post-induction hypotension. Prompt management for subsequent hypotension should be considered if a patient with hypertension has a PI of the ear ≤ 0.96 prior to the induction of anesthesia. Further studies are required to establish the usefulness of the PI of the ear.

## Figures and Tables

**Figure 1 jcm-11-06342-f001:**
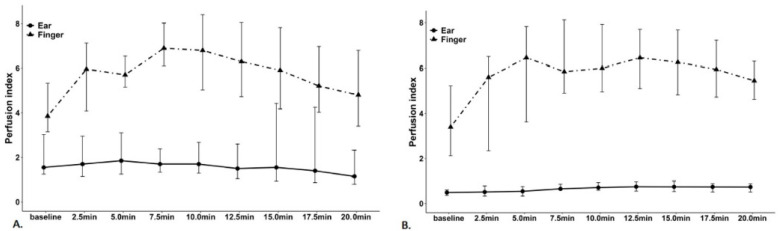
The variability in the ear and finger perfusion indices during the post-induction period in the non-hypotension and hypotension groups: (**A**) Non-hypotension group. (**B**) Hypotension group.

**Figure 2 jcm-11-06342-f002:**
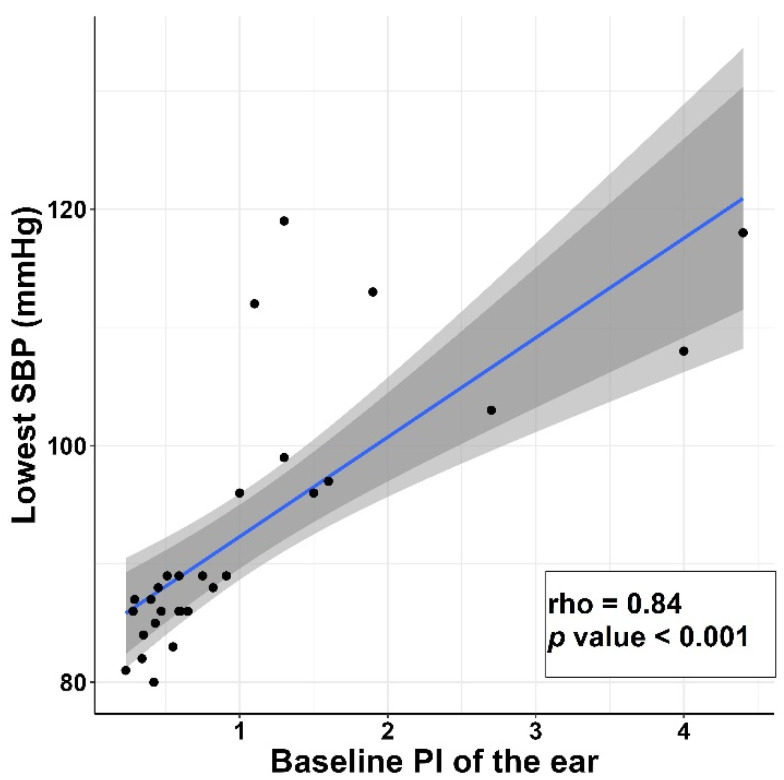
The linear correlation between the baseline PI of the ear and the lowest SBP (mmHg) of the patients. PI, perfusion index; SBP, systolic blood pressure.

**Figure 3 jcm-11-06342-f003:**
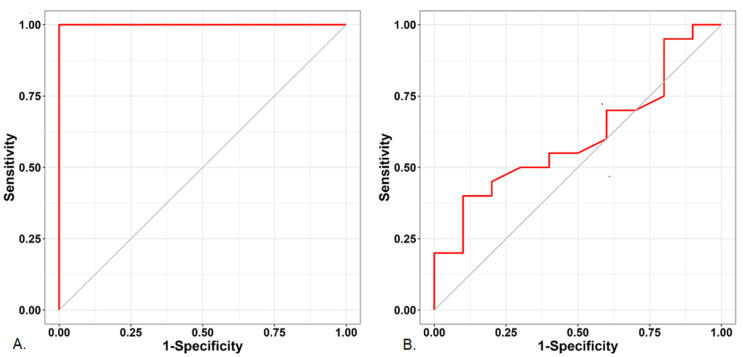
Comparing the receiver operating characteristic curves of the perfusion indices of the (**A**) ear and (**B**) finger.

**Table 1 jcm-11-06342-t001:** Demographic data and baseline characteristics (*n* = 30).

Variable	Subgroup	Non-Hypotension Group (*n* = 10)	Hypotension Group (*n* = 20)	*p*-Value
Sex				0.440
	Male	7 (70)	10 (50)	
	Female	3 (30)	10 (50)	
Age (years)		68 (66.75–73.25)	65.5 (58.75–71.75)	0.074
Height (cm)		166.5 (162.75–170)	159.5 (151–166.5)	0.058
Weight (kg)		66 (61.5–75.25)	64.5 (59.25–71.5)	0.441
BMI (kg/m^2^)		24.74 (22.27–27.89)	24.72 (22.86–28.74)	0.914
Underlying disease				0.690
	HTN	6 (60)	14 (70)	
	HTN/DM	4 (40)	6 (30)	
Propofol dose (mg)		70.85 (65.35–79.23)	70.25 (62.43–80.15)	0.509

Continuous variables are presented as medians (interquartile ranges), and *p*-values were computed using the Wilcoxon rank-sum test. Categorical variables are presented as sample numbers (%), and *p*-values were calculated using Fisher’s exact test. BMI, body mass index; HTN, hypertension; DM, diabetes mellitus.

**Table 2 jcm-11-06342-t002:** Comparison of baseline hemodynamic data between patients with and without hypotension after the induction of anesthesia.

Variable	Non-Hypotension Group (*n* = 10)	Hypotension Group (*n* = 20)	*p*-Value
SBP (mmHg)	142.5 (135.5–165.25)	142.5 (131.5–156)	0.403
DBP (mmHg)	79.0 (76–88.5)	76.5 (70–88.75)	0.692
MBP (mmHg)	96.5 (92.75–109)	97.0 (91.5–103.75)	0.843
HR (beats/min)	63.5 (58–76)	67.5 (59.75–73)	0.377
Finger			
PVI	11.0 (6.75–22.75)	10.0 (8–14.75)	0.566
PI	3.85 (3.15–5.33)	3.4 (2.12–5.22)	0.355
Ear			
PVI	15.0 (11.75–20.75)	18.0 (14–21)	0.354
PI	1.55 (1.25–3.03)	0.49 (0.36–0.6)	<0.001 *

Continuous variables are presented as medians (interquartile ranges), and *p*-values were computed using the Wilcoxon rank-sum test. SBP, systolic blood pressure; DBP, diastolic blood pressure; MBP, mean blood pressure; HR, heart rate. PVI, pleth variability index; PI, perfusion index; * *p* < 0.05.

**Table 3 jcm-11-06342-t003:** Correlation between the lowest SBP and hemodynamic variables measured at baseline.

Variable	Rho (95% CIs)	*p*-Value
SBP (mmHg)	0.062 (−0.306–0.413)	0.745
DBP (mmHg)	0.211 (−0.165–0.534)	0.262
MBP (mmHg)	0.108 (−0.264–0.451)	0.571
HR (beats/min)	−0.071 (−0.421–0.297)	0.710
Finger		
PVI	0.231 (−0.146–0.549)	0.219
PI	0.331 (−0.043–0.624)	0.074
Ear		
PVI	−0.243 (−0.558–0.134)	0.196
PI	0.839 (0.652–0.930)	<0.001 *

Rho (95% CIs), Spearman’s correlation coefficient and 95% Cis; CI, confidence interval; SBP, systolic blood pressure; DBP, diastolic blood pressure; MBP, mean blood pressure; HR, heart rate; PVI, pleth variability index; PI, perfusion index; * *p* < 0.05.

**Table 4 jcm-11-06342-t004:** Comparison of AUCs for preoperative hemodynamic variables to predict post-induction hypotension.

Variable	AUC (95% CIs)	*p*-Value	Sensitivity	Specificity	Cutoff
SBP (mmHg)	0.597 (0.376–0.819)	0.391	0.35	0.9	≤133.5
DBP (mmHg)	0.548 (0.330–0.765)	0.676	0.50	0.8	≤76
MBP (mmHg)	0.525 (0.293–0.757)	0.826	0.90	0.3	≤106.5
HR (beat/min)	0.602 (0.362–0.843)	0.367	0.95	0.3	≥58.5
Finger					
PVI	0.568 (0.318–0.817)	0.553	0.95	0.3	≤20
PI	0.608 (0.396–0.819)	0.344	0.40	0.9	≤2.75
Ear					
PVI	0.608 (0.368–0.847)	0.344	0.95	0.3	≥12.5
PI	1.000 (1.000–1.000)	<0.001	1.00	1.0	≤0.96

AUC (95% CIs), AUC and 95% CIs; CI, confidence interval; SBP, systolic blood pressure; DBP, diastolic blood pressure; MBP, mean blood pressure; HR, heart rate; PVI, pleth variability index; PI, perfusion index; AUC, area under the curve; Cut-off, cut-off to predict hypotension, which is selected at the point maximizing Youden’s J statistic.

## Data Availability

The datasets used and analyzed during the present study are available from the corresponding author upon reasonable request.

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
