# Peer review of "The Perfusion Index of the Ear as a Predictor of Hypotension Following the Induction of Anesthesia in Patients with Hypertension: A Prospective Observational Study"

_jcm, 2022, doi:10.3390/jcm11216342_

Round 1

Reviewer 1 Report

It is an interesting study and has some supporting scientific background. However, there are some major concerns that the authors should consider. Also, some improvements are needed in the manuscript.

Major.

1. Lack of generalizability

In this study, no opioid was administered during the observation period. However, as balanced anesthesia (hypnotics + opioid) is the common practice, the result of the current study is limited by its generalizability.

2. Definition of hypotension

Although there are various definitions of hypotension exist across studies, mean arterial pressure (MAP) is a more dominantly used standard. Also, MAP is a more relevant parameter for organ perfusion and clinical outcomes. In this regard, predicting systolic pressure <90mmHg is a less clinically relevant finding than MAP is. How about performing additional exploratory analysis with hypotension defined with MAP?

3. Lack of mechanism of hypotension.

The authors repeatedly stated the relationship between vasodilation, post-induction hypotension, and perfusion index of the ear. However, there were no significant changes in the ear PI during the monitoring period. Thus, the baseline PI value and the following (relatively unchanged) ear PI values were the relevant parameters for post-induction hypotension. In this perspective, postinduction vasodilation was not represented in the ear PI.

On the other hand, the decrease in vascular tone (vasodilation) seems to partially be represented in the finger PI (increase in finger PI). However, it seems that finger PI values have no predictive power for post-induction hypotension. In this logic, there is no clear relationship between the change in peripheral vasodilation and post-induction hypotension.

According to the authors' assumption that ear PI represents central vascular tone, only the baseline central vascular tone can explain the phenomenon described in the study.

Please interpret the results carefully and revise accordingly.

Minor

1. line 45-49. Although the meaning of the sentence is delivered, it is not readily comprehended. Recommend rephrasing this sentence. (Increase readability)

2. Reference: The format of the reference should be checked (duplicated numbering)

3. line 50-53. Recommend adding a citation regarding the described issue. I understand that the issue is a natural insight for an anesthesiologist, but some readers may not be familiar with this issue and may need a reference to inquire about a more detailed explanation.

4. Pi vs PI: recommend changing the abbreviation of the perfusion index from Pi to PI. PI seems to be a more dominantly used abbreviation. Is there a reason to use Pi rather than PI?

5. line 64-68. The cited studies do not seem to guarantee the superiority of ear Pi over digit Pi. Authors should suggest a more robust rationale (i.e., supporting studies) regarding this issue since it is a central theme of the current study. Otherwise, this issue should be stated as a hypothesis rather than a fact.

6. Specification of the site of monitoring. Earlobe or tragus or ear canal? Please describe more detailed site description.

7. line 70-71. This study does not include a measurement process of central vascular status. Thus, the relationship between ear PI and postinduction hypotension does not directly implicate a relationship between ear pi and central vascular status, despite the logic being probable.  

6. line 76. “The Institutional Review Board approved this study (IRB) and the Hospital Research Ethics Committee”

Please re-check this sentence. The hospital research ethics committee has no corresponding verb in this sentence.

7.  Exclusion criteria: how did the ‘autonomic nervous system impairment’ determined?

8. line 92. It is more prudent to describe PI first rather than PVI since the former is the primary concern in the current study.

9. Line 80-83.

The registration of the first patient does not match the described enrollment.

CRIS: 2022.4.7~2022.7.7

10. Table 1.

p<0.0001”. The commonly agreed standard for p-value description for a low value is “<0.001”

11. line 150-151. This sentence seems lost its proper position.

12. line 158-160. These sentences better be integrated into a single sentence.

13. line 164. Check the rho value. Negative?

14. line 165. “the regression equation between the two variables was shown as….??”

15. line 212. “mainly causes” >> Decrease in SVR is the main cause (or mechanism) of hypotension

16. line 222. There is no “Awad” in the cited reference (18)

17. line 242-243. “intense artery” seems to be a non-standard description. Please consider other terminology.

18. line 261-267. The contrast between citation 12 and the current study was not clearly revealed in the paragraph. The differences in the inclusion/exclusion criteria should be adequately discussed.

Also, “with hypertension” in lines 262 and 263 seems duplicated.

19. line 268-272. I do not agree that a “small sample size” per se is a limitation of a study. The sample size should be determined based on proper statistics, and there is no justification for recruiting patients beyond the determined size.

20. Conclusion

A conclusion should focus on the primary outcome (primary hypothesis) and be concise. Avoid stating beyond the result of the current study.

This study does not include critically ill patients. Thus, the generalizability of the result to the patients deviated from the current study has never been demonstrated. Stating applicability to the critically ill patient is an overstatement.

Author Response

Thank you for your constructive comments. It helps us considerably improve the manuscript and enhance its clarity. We have responded to each of your comments in a point-by-point manner. Please see the attachment. In addition, we have provided a revised version of the manuscript with the changes marked. We hope that the changes made to the revised manuscript satisfactorily address your concerns.

Author Response

(The authors gave the same response as above.)

Reviewer 3 Report

20 Some definition of Pi would be good.  Most physicians have no idea what this means.

25-29 Three sentences say the same thing.  One suffices.

52-53 ?reference

60  Again, define Pi.  Specifics please.

65-68 This sentence does not make sense.  Maybe the authors are stating that ear perfusion more closely reflects central perfusion than digital perfusion, and therefore is more informative of central organ ischemia.  But “stable” is probably the wrong way to phrase it.

83-85 It seems to me that one of the factors in determining blood pressure response to anesthetic vasodilation is the preoperative LV volume status, which is related to LV compliance.  As all of these patients, had hypertension which is a known precipitant, how many of the hypotensive events occurred in patients with LVH?  This would necessitate a preop TTE but at least worth commenting in the discussion. 

92 Define PVI and Pi please

94 Mean arterial blood pressure is usually abbreviated as MAP, not MBP

103-105  None of these patients were intubated.  This is worth noting (also in the abstract), especially since the stimulation of laryngoscopy and ETT insertion often corrects post-induction hypotension.

145-147 What were the other hypertensive meds?  Was there a correlation between med type and postinduction  hypotension (e.g., ARBs or ACE inhibitors)?

148  Figure 1 is probably unnecessary.

152 Table 1 is information dense.  Simplify it please.

159-165  This is confusing and appears repetitive.  The word usage and grammar needs alteration.

All figures and tables:  State what the figure or table is supposed to show.  One sentence.  For many of these items, it is NOT clear.

185+ Syntax needs work

The second paragraph in Discussion is written in a confusing and awkward manner.  For example (215), the authors mean that that propofol was injected less rapidly, but they fail to use any adverb.  This is one of many examples in which this paper fails to communicate.

227 In addition, proximity to the brain ensures stable perfusion solely through the 227 external carotid artery” needs a reference.

234+  This is entirely conjecture (as the authors note).  Consider other possibilities related to LV compliance, drugs, etc.

251 Your hypotheses can be adjusted for stroke volume considerations.  See above.

Author Response

(The authors gave the same response as above.)

Round 2

Reviewer 1 Report

Line 5, Affiliations should be indicated using superscript

Line 20, “could be shown”  à can show?

Line 22, “the ear is more reliable à is this a confirmed fact or a probable hypothesis? I think it is not a robust fact yet. Consider rephrasing.

Line 23, 91-92, “we hypothesized that PI measured in the ear could represent the preoperative hemodynamic status in patients with hypertension” à although it is a probable explanation of the underlying mechanism of the result, a more direct hypothesis that this study has dealt with would be “could preoperative ear PI be a predictive parameter for postinduction hypotension in hypertensive patients?”. Please reconsider. (or combined the underlying mechanism and hypothesis, for example, “~could represent the preoperative hemodynamic status ,and thus, predict post-induction hypotension”)

Line 59, completed à “subsided” would be a more proper term.

Line 68-69, recommend removing (duplicate with the former sentence), or revising as, for example, “determining preoperative hemodynamic status can be considered a cardinal aspect of management of patients with hypertension”

Line 113, Masimo Radical 7 is a monitor, not a sensor. Please describe the specific sensors used for the finger and ear, respectively. (maybe E1 ear sensor for the ear?)

Line 192-3, what does it mean “hypotension occurred at 15 min from baseline in the hypotension group”. Did all hypotensive occurr at 15 min postinduction? It is unlikely all the onset of hypotensive events were the same.

Line 200, “and in the non-hypotension group~”.  “and” better be substituted by “whereas”

Line 208, again, check the rho value, negative? (it is correctly stated as positive value in the figure and the table)

Line 208-9, “and was a linear correlation in the linear regression analysis”. Please correct the sentence.

Overall, I recommend extensive English editing. 

Author Response

(The authors gave the same response as above.)

Reviewer 2 Report

Line 85 and 268 - the end of the sentence "from the blood and from blood to quantity" -  the formulations are unclear

Line 91 - in the ear - please correct - to ear probe

Line 325 - non hypertensive - correct to non hypotensive

Author Response

(The authors gave the same response as above.)

Reviewer 3 Report

The paper is significantly improved. 

The English language is still awkward with shifting between present and past tense within one paragraph, repeated use of the verb "can" (often in past tense), and incorrect choice of verbs.  Adjacent sentences are sometimes repetitions  of one another.  Altering this presentation will go a long way to making the paper more readable.

There are numerous instances of these errors in the paper.   For example: We hypothesized that PIPi measured in the ear could represent preoperative hemodynamic status in patients with hypertension. The preoperative PI of the ear could help predict post-induction hypotension-related vasodilation in patients with hypertension. 

These two sentences should be combined to state:  We hypothesized that low values for PI measured in the ear are predictive of post-induction hypotension.

Author Response

(The authors gave the same response as above.)
